# From Serendipity to Rational Identification of the 5,6,7,8-Tetrahydrobenzo[4,5]thieno[2,3-*d*]pyrimidin-4(3*H*)-one Core as a New Chemotype of AKT1 Inhibitors for Acute Myeloid Leukemia

**DOI:** 10.3390/pharmaceutics14112295

**Published:** 2022-10-26

**Authors:** Andrea Astolfi, Francesca Milano, Deborah Palazzotti, Jose Brea, Maria Chiara Pismataro, Mariangela Morlando, Oriana Tabarrini, Maria Isabel Loza, Serena Massari, Maria Paola Martelli, Maria Letizia Barreca

**Affiliations:** 1Department of Pharmaceutical Sciences, “Department of Excellence 2018–2022”, University of Perugia, 06123 Perugia, Italy; 2Hematology and Clinical Immunology, Department of Medicine and Surgery, University of Perugia, 06132 Perugia, Italy; 3CIMUS Research Center, University of Santiago de Compostela, 15782 Santiago de Compostela, Spain

**Keywords:** AKT1, kinase inhibitors, cancer, acute myeloid leukemia, rare disease, molecular modeling, molecular docking, similarity search, MD simulation, computer-aided drug discovery

## Abstract

Acute myeloid leukemia (AML) is a heterogeneous hematopoietic malignancy whose prognosis is globally poor. In more than 60% of AML patients, the PI3K/AKTs/mTOR signaling pathway is aberrantly activated because of oncogenic driver alterations and further enhanced by chemotherapy as a mechanism of drug resistance. Against this backdrop, very recently we have started a multidisciplinary research project focused on AKT1 as a pharmacological target to identify novel anti-AML agents. Indeed, the serendipitous finding of the in-house compound **T187** as an AKT1 inhibitor has paved the way to the rational identification of new active small molecules, among which **T126** has emerged as the most interesting compound with IC_50_ = 1.99 ± 0.11 μM, ligand efficiency of 0.35, and a clear effect at low micromolar concentrations on growth inhibition and induction of apoptosis in AML cells. The collected results together with preliminary SAR data strongly indicate that the 5,6,7,8-tetrahydrobenzo[4,5]thieno[2,3-*d*]pyrimidin-4(3*H*)-one derivative **T126** is worthy of future biological experiments and medicinal chemistry efforts aimed at developing a novel chemical class of AKT1 inhibitors as anti-AML agents.

## 1. Introduction

The AGC kinases AKT1-3 are key mediators of the PI3K/AKT/mTOR signaling pathway, which plays a crucial role in major cellular functions such as proliferation, migration, and antiapoptotic survival [1]. The main downstream targets of AKT include the mammalian target of rapamycin (mTOR) of the mTOR complex 1 (mTORC1), that through phosphorylation activates the p70 ribosomal S6 kinase (p70S6K) and inactivates the 4E-BP1 (an inhibitor of the elongation initiating factor 4E, eIF4E) ultimately promoting protein translation and cell growth [2]. Among the three AKT isoforms, AKT1 is expressed ubiquitously and a high rate of AKT1 amplification occurs in a wide range of human cancers [3] including acute myeloid leukemia (AML) [4,5], a rare but often fatal blood cancer. The major activators of the PI3K/AKT/mTOR pathways are upstream kinases, such as tyrosine kinase receptors (e.g., fms-like tyrosine kinase 3, FLT3) or K/NRAS frequently involved in gain-of-function gene mutations in AML [6]. Indeed, a constitutive activation of the AKT pathway and the phosphorylation of its targets such as mTOR, p70S6K, and 4E-BP1 are detected in about 60% of AML patients [7,8,9], and are mostly described as connected with a poorer prognosis [10,11,12,13,14]. The level of AKT1 expression and kinase activity is often associated with the degree of differentiation, hormone dependency, and aggressive behavior of cancer cells and/or with a less favorable prognosis [12,13,14]. Additionally, accumulating evidence indicates that the acquisition of resistance to chemotherapeutic drugs involves the activation of the PI3K/AKT/mTOR pathway [15], suggesting that AKT1 inhibitors could be useful for overcoming drug resistance and improving response [16,17]. Recently, AKT activation in AML has also been implicated as a resistance mechanism to the targeted therapy **selinexor**, a newly FDA-approved small molecule inhibitor of exportin 1 (XPO1) [18].

In the era of precision oncology, AKT1 thus represents an attractive target for the development of targeted therapies selectively hitting cancer cells characterized by an aberrant activation of the PI3K/AKT/mTOR signaling pathway. To date, no drugs against AKT1 are on the market, although some AKT1 inhibitors are currently in clinical trials for solid cancers [19,20], but not for AML, which remains an urgent medical need. Against this backdrop, we have very recently started a multidisciplinary research program aimed at rationally identifying new AKT1 inhibitors as anti-AML agents.

## 2. Materials and Methods

### 2.1. Computational Studies

#### 2.1.1. Ligand Preparation

All compounds explored in molecular modelling studies (i.e., **T187**, the clinical candidates, and the selected UNIPG-lib compounds) were prepared using the LigPrep [21] to produce low energy 3D structures taking into account the ionization states, stereochemistry, the tautomerism, and ring conformations at the desired pH (7.5 ± 0.5).

#### 2.1.2. Similarity Search

The BioSolveIT Feature Tree (FTrees) [22] software tool was used to perform fast 2D-similarity assessments. The similarity searches were performed using the following command line:

ftrees -i T187.sdf -l $library –minSimilarityThreshold X -o output.csv

where the -i option defines the query molecule, the -l option states the screened library, the –minSimilarityThreshold defines the similarity threshold below which molecules are discarded, and the -o option states the output file.

To compute the similarity between **T187** and the known AKT1 inhibitor clinical candidates, the compounds reported in Appendix A were used as the $library file and the –minSimilarityThreshold was set to 0.

To build the **T187**-like compounds’ library, the prepared UNIPG-lib was used as the $library file and the -minSimilarityThreshold was set to 0.65.

#### 2.1.3. Protein Preparation

The only two crystal structures of AKT1 co-crystallized with clinical candidates (i.e., **capivasertib** and **ipatasertib**) available from RCSB PDB [23] were PDB ID 4GV1 [24] and 4EKL [25], respectively. Considering the best resolution of the 4GV1 structure (1.49 Å vs. 2.00 Å for 4EKL), this protein conformation was used for further modeling studies. Firstly, the complex was prepared using Schrödinger’s Protein Preparation Wizard (Schrödinger Release 2019-2) [26] in order to obtain satisfactory starting structures for the docking studies. The complex was pre-processed as follows: (i) hydrogen atoms were added and bond orders were assigned to amino acid residues and ligands, (ii) the missing side chains were filled, (iii) the protein was capped with acetyl (ACE) and N-methylamine (NMA) groups, (iv) all water molecules beyond 3 Å from heteroatoms were deleted, and (v) Epik (Schrödinger Release 2019-2) [27] was used to predict the ionization and tautomeric states for the ligands (pH = 7.0 ± 2). In addition, the structure presented some missed residues (loop 451–457) that were filled by using the homology modelling tool function implemented in the Prime package (Schrödinger Release 2019-2) [28]. Then, the H-bond network of the complex was optimized using PROPKA for the assignment of the residue protonation states (pH = 7.0). Finally, the complex was submitted to a restrained minimization (OPLS3 force field), which was stopped when the RMSD of heavy atoms reached 0.30 Å.

#### 2.1.4. Molecular Docking

Docking studies were performed using Schrödinger’s Glide software (Schrödinger Release 2019-2) [29]. In our protocol, the crystallographic position of **capivasertib** (PDB ID 4GV1) was used as reference to center the grid on the ATP-site. The docking space was defined as a cubic box (28 Å outerbox), with an inner cubic box (14 Å) defining the region where the centroid of the ligand had to be located. The centroid for the 4GV1 protein had coordinates of −20.00, 4.33, and 11.27 to the *x*-, *y*-, and *z*-axis, respectively. After grid preparation with the “receptor grid generation” tool, docking experiments were performed using the Glide XP (Extra Precision) protocol (Schrödinger Release 2019-2) [30], by generating three poses for each ligand. The generated poses were submitted to the MMGBSA minimization step using the “Refine Protein-Ligand Complex” tool in Prime (default settings—esidues at 5 Å within the ligand considered as flexible) and only the pose with the lower MM-GBSA ∆G of binding (MM-GBSA dG Bind) was retained.

The collected binding poses were then filtered for the presence of H-bond interaction(s) with the hinge residues Ala230 and/or Glu 228.

By using the above-described protocol, the initial hit, **T187,** was predicted to bind the catalytic site with a MM-GBSA dG Bind of −39.42 Kcal/mol. Based on this result, the **T187**-like compounds were docked and later filtered using a threshold of −40 Kcal/mol on the MM-GBSA dG Bind and then submitted to visual inspection.

The two co-crystallized advanced inhibitors were added to the ligand set for comparison. The best docking poses of **ipatasertib** (MM-GBSA dG Bind = −67.72 Kcal/mol) and **capivasertib** (MM-GBSA dG Bind = −62.42 Kcal/mol) agreed well with their experimental binding conformations (Appendix A).

#### 2.1.5. Molecular Dynamics Simulations

Explicit solvent Molecular Dynamics (MD) simulations were performed by using the Desmond package v5.8 (Schrödinger Release 2019-2) [31] to investigate the stability of the MMGBGA binding modes generated for **T126** and **T125** in complex with the 4GV1 protein conformation. The OPLS3e forcefield was selected and the systems were solvated in an orthorhombic box (size = 20.0 × 20.0 × 20.0) exploiting the TIP3P water molecule mode. Na^+^ and Cl^−^ counter ions at a concentration of 0.15 M were added to neutralize the system charge by means of the System Builder tool. The system setup step enabled the generation of the 4GV1-**T126** and 4GV1-**T125** complex systems consisting of 83,768 and 83,767 atoms, respectively. MD simulations were set up through the Molecular dynamics tool. The systems were relaxed before the simulation by using the protocol implemented in Desmond which comprises 5 equilibration steps. The system minimization was followed by a production step of 200 ns that was simulated by the NTP ensemble at 310 K and 1 atm. For each system, three replicas with different random seeds were performed. MD simulation results were analyzed through the Simulation interaction diagram (SID) tool provided by Schrödinger.

### 2.2. Chemistry

#### 2.2.1. Synthetic Procedures

All the compounds evaluated in the biological assays (Appendix A) were synthesized as previously reported by us, with the exception of compound **T187** which has never been reported so far and, therefore, is described in this paper, and compounds **T124**, **T126,** and **T128** that were synthesized as reported in the literature [32,33,34,35].

Compound **T187** was synthesized as reported in Figure 1, starting from a one-pot Gewald reaction entailing the reaction of cycloheptanone with cyanoacetamide in the presence of sulphur and morpholine in EtOH at reflux, to give the intermediate 1 [36]. Successive amidation of compound **1** [36] with 2,4-difluorobenzoic acid was performed in CH_2_Cl_2_ at 50 °C, in the presence of the Mukaiyama reagent (2-chloro-1-methylpyridinium iodide, CMPI), 4-dimethylaminopyridine (DMAP), and triethylamine (TEA), furnishing compound **T187**.

#### 2.2.2. Experimental Section

The commercially available starting materials, reagents, and solvents were used as supplied. All reactions were routinely monitored by TLC on silica gel 60F254 (Merck) and visualized by using UV or iodine. Flash column chromatography was performed on Merck silica gel 60 (mesh 230–400). After extraction, the organic solutions were dried over anhydrous Na_2_SO_4_, filtered, and concentrated with a Büchi rotary evaporator at reduced pressure. Yields are of the purified product and were not optimized. The purities of compounds **T124**, **T126**, and **T128** (96.5%, 100%, and 100%, respectively) were determined by HPLC analysis using a Jasco LC-4000 instrument equipped with a DAD Jasco MD-4015 detector from 200 to 600 nm. The purity was revealed at 344 nm using an XTerra MS C18 Column, 5 µm, 4.6 mm × 150 mm with a flow rate: 0.4 mL/min; an acquisition time: 10 min; an isocratic: acetonitrile containing 0.1% of formic acid (85%) and water (15%); a column temperature: 25 °C; and an injection volume: 20 μL. The peak retention time is given in minutes and the Chromatograms were analyzed by Chromatography Data System software ChromNAV version number 2.03.06 [Build 4] (2003 JASCO Corporation, Tokyo, Japan). The purity of compound **T187** (95.09%) was determined by LC/MS using an Agilent 1290 Infinity System machine equipped with a DAD detector from 190 to 640 nm. The purity was revealed at 254 nm using a Phenomenex AERIS Widepore C4, 4.6 mm, 100 mm (6.6 lm) with a flow rate: 0.6 mL/min; an acquisition time: 10 min; a gradient: acetonitrile in water containing 0.1% of formic acid (0 100% in 10 min); a column temperature: 40 °C; and an injection volume: 2 μL. The peak retention time is given in minutes. Accurate mass measurements were performed with HRMS (Agilent Q-TOF 6540) and the monitored *m*/*z* values ranged from 100 *m*/*z* to 3000 *m*/*z* in positive polarity. The Mass spectrometry was equipped with an ESI source. The gas temperature, gas flow, nebulizer, sheath gas, and sheath gas flow were set at 350 °C, 9 L/min, 35 psig, 400 °C, and 9 L/min, respectively. The capillary voltage was set at 4000 V and the fragmentor at 120 V. The 1H NMR and 13C NMR spectra were recorded on a Bruker Avance DRX-400 MHz using a residual solvent such as dimethylsulfoxide (δ = 2.48) as an internal standard. The chemical shifts were recorded in ppm (δ) and the spectral data are consistent with the assigned structures. The spin multiplicities are indicated by the: d (doublet), t (triplet), and m (multiplet).

2-(2,4-Difluorobenzamido)-5,6,7,8-tetrahydro-4*H*-cyclohepta[*b*]thiophene-3-carboxamide (**T187**). 2,4-Difluorobenzoic acid (0.09 g, 0.61 mmol), CMPI (0.14 g, 0.57 mmol), DMAP (0.03 g, 0.24 mmol), and TEA (0.14 g, 1.4 mmol) were added to a solution of compound **1 [36]** (0.10 g, 0.47 mmol) in dry CH_2_Cl_2_ (5 mL). The reaction mixture was maintained at 40 °C overnight and then, after colling, it was evaporated to dryness. The obtained residue was suspended in a 1 M HCl solution, stirred for 10 min, and then filtered. The solid was purified by flash chromatography eluting with CHCl_3_/MeOH (98:2), to give **T187** (0.07 g, 42%). ^1^H-NMR (400 MHz, DMSO-*d*_6_) δ 1.55–1.65 (m, 4H, cycloeptane CH_2_), 1.76–1.80, 2.70–2.75, 2.78–2.80 (m, each 2H, cycloheptane CH_2_), 7.28–7.32 (m, 1H, aromatic CH), 7.47–7.70 (m, 3H, aromatic CH and NH_2_), 8.00–8.10 (m, 1H, aromatic CH), 11.66–1168 (d, *J* = 8.5 Hz, 1H, NH); ^13^C-NMR (101 MHz, DMSO-*d*_6_) δ 27.7, 28.1, 28.7, 28.8, 32.1, 105.6 (t, *J_C_*_–*F*_ = 27.3 Hz), 113.3 (d, *J_C_*_–*F*_ = 20.2 Hz), 117.6 (d, *J_C_*_–*F*_ = 9.1 Hz), 122.1, 131.6, 133.8 (d, *J_C_*_–*F*_ = 11.1 Hz), 135.6, 137.2, 158.7, 159.6, 160.8 (dd, *J_C_*_–*F*_ = 14.1 and 253.5 Hz), 165.0 (dd, *J_C_*_–*F*_ = 13.1 and 266.6 Hz), 167.9. HRMS: *m*/*z* calcd for C_17_H_16_F_2_N_2_O_2_S 351.0089 (M + H)^+^, found 351.0978 (M + H)^+^. HPLC, ret. time: 5.633 min, peak area: 95.09%.

### 2.3. Enzymatic Assays

The studied compounds, including the standard inhibitor Staurosporine (Thermo-Fisher , Madrid, Spain, Invitrogen BP2541), were incubated with 3 nM AKT1 (#Aps 01-101; Kinase Logistics, Meyn, Germany) and 1.5 μM Profiler Pro peptide 6 (#760350; Perkin Elmer, Tres Cantos, Madrid, Spain) in the presence of different concentrations of Adenosine 5′triphosphate disodium salt, ATP (#A2383; Sigma Aldrich, Madrid, Spain) in a 384-well Microplate (#781076; Greiner, San Sebastián de los Reyes, Madrid, Spain). The reaction rate was calculated from kinetic measurements carried out measured in a Caliper EzReader LabChip 3000 (Caliper, Hopkinton, MA, USA) reader with samples taken each 10 min for a total time of 60 min. The percentage of substrate conversion rate was plotted versus the ATP concentration. The data was analyzed by non-linear fitting with GraphPad Prism by employing the Michaelis–Menten equation. The buffer assay used in the inhibition hAKT1 assay is: Hepes 100 mM, DMSO 4%, Brij 0.003%, Tween 20 0.004%, MgCl_2_ 10 mM, with a pH = 7.5.

The K_i_ values were calculated by employing the GraphPad Prism equations.

### 2.4. Cell-Based Assays

#### 2.4.1. Cells and Compound Treatments

The human AML cell lines, OCI-AML3 (carrying the *NPM1* gene mutation A and the *DNMT3A*^R882C^ mutation), IMS-M2 (carrying the *NPM1* gene mutation A), OCI-AML2 (with the wild-type *NPM1* and *DNMT3A*^R635W^ mutation), and SKM-1 and MOLM-13 (with the wild-type *NPM1*) were previously reported [37,38]. The cell lines were purchased from the Deutsche Sammelung von Mikroorganismen und Zellkulturen (DSMZ) (Braunschweig, Germany) cell bank and maintained according to the vendors’ instructions. Briefly, cultured cells were split every 3 days and maintained in an exponential growth phase. The cell lines were tested for mycoplasma using the Universal Mycoplasma Detection Kit (ATCC). The OCI-AML3 were grown in α-MEM supplemented with 20% fetal bovine serum, 100 U/mL penicillin, and 100 μg/mL streptomycin (P/S), and all other cell lines were cultured in RPMI and supplemented with a 10–20% fetal bovine serum and 100 U/mL penicillin and 100 μg/mL streptomycin. The cells were kept at 37 °C in a 5% CO_2_ incubator. Prior to screening, the appropriate cell concentration for all cell lines was determined. For the cell lines’ treatments, cells were maintained as follows: OCI-AML3 at 0.25 × 10^6^/mL, OCI-AML2 and IMS-M2 at 0.35 × 10^6^/mL, and SKM-1 and MOLM-13 at 0.5 × 10^6^/mL. The compounds were re-suspended at 10 mM stock in DMSO. Further dilutions were performed in phosphate-buffer-saline (PBS) for pharmacological experiments. The cells were exposed to compounds at different concentrations for the indicated time points and sampled as described below. Briefly, the cells (4 × 10^5^/mL) were seeded in 384-well plate and treated for 48 h with various concentrations (0.001 μM to 100 μM) of either one of the compounds, **T101**, **T126,** and **T159,** or **NSC348884**, as a positive control [39]. Each plate included DMSO at 0.1% as a negative control. The metabolic cell activity was measured by using the CellTiter Blue proliferation agent (Roche) as described in the manufacturer’s procedure. Briefly, CellTiter Blue solution (Resazaurin) was added to each well and the plates were incubated at 37 °C with 5% CO_2_ for 4 h. During the incubation time, the resazurin is reduced to fluorescent resorufin in metabolically and proliferating living cells. The fluorescence intensity was measured using the automated Tecan plate reader (Tecan Group Ltd., Männedorf, Switzerland) at an excitation/emission wavelength of 531/572 nm. The fluorescence readouts were reported as relative fluorescence units (RFU) and read on a luminescence plate reader (Infinite 200 Pro, Tecan, CA, USA). The dose response curves (maximal inhibitory concentration (IC_50_)) were obtained by non-linear regression analysis using the GraphPad Prism 5.0 program. The concentration is shown as a logarithmic function. All the experiments were performed at least in triplicate, with technical repetition. The results are expressed as mean ± standard error of the mean (SEM). For the Colony Forming Unit (CFU) assay, 10 × 10^6^ CD34+ human hematopoietic progenitors were received from unique healthy donors, upon signed informed consent, and were cultured in SFEM II (Serum-free medium) and P/S at a density of 1 × 10^6^/mL with cytokines that are important for cell division and self-renewal: fms-like tyrosine kinase 3 (FLT3)-ligand, FLT3L (100 ng/mL), Stem Cell Factor, SCF (100 ng/mL) and Trombopoietin, and TPO (50 ng/mL) for 48 h before plating them on a Methocult medium.

#### 2.4.2. Colony Forming Unit (CFU) Assay

For the CFU assay, 500 CD34+ human cells were plated in a MethoCult™ enriched medium (see Appendix A) on a 6-well plate (untreated cells versus **T126** at 1, 2.5, and 5 µM, each condition in triplicate). The plates were then incubated for 14 days at 37 °C (5% CO_2_), and the images were captured and the colonies counted by STEMvision™. STEMvision™ is a bench-top instrument and computer system designed specifically for the automated imaging and counting of hematopoietic colonies in the CFU assay. This system has been optimized for use with MethoCult™ media and meniscus-free SmartDish™ culture ware. The use of this standardized platform significantly improves the accuracy and reproducibility of the human CFU assay. Mobilized peripheral blood (MPB) analysis packages have been used to accurately count colonies.

#### 2.4.3. Cell Lysates Preparation and Western Blot Analysis

Immunoblotting was conducted as described previously [37]. Briefly, fresh cell pellets (0.2–0.5 × 10^6^/test for AML cell lines) were dissolved directly in 30–60 μL of Laemmli sample buffer 1× (1.5 M Tris-HCl pH 6.8, glycerol, β-mercaptoethanol, SDS, bromophenol blue) and boiled at 95 °C for 5 min. The proteins were separated by SDS-polyacrylamide gel electrophoresis (SDS-PAGE) on Precast 4–15% gradient gels (Biorad, Hercules, CA, USA) transferred onto nitrocellulose membranes (GE Healthcare, Piscataway, NJ, USA) and probed with specific primary antibodies followed by horseradish peroxidase-conjugated secondary antibodies (GE Healthcare Lifesciences). The polypeptides were visualized using enhanced chemiluminescence (Luminata Crescendo, Merck Millipore, Burlington, MA, USA) according to the manufacturer’s instructions. The bands were visualized using the Biorad Chemidoc and the images were processed using the image lab software (Biorad) and Adobe Photoshop (Ps CC 2019). β-actin expression levels were used as a control for protein loading, as indicated.

#### 2.4.4. Cell Growth and Apoptosis Evaluation

The viable cell count was evaluated daily using the Invitrogen Countess automated cell counter (Invitrogen, Carlsbad, CA, USA) by trypan blue exclusion, and individual growth curves were generated from these data.

The cell apoptosis was assessed using Annexin V APC- or FITC-conjugated antibodies and counterstained with Propidium Iodide (PI) or 7AAD for the detection of necrotic cells (Becton Dickinson, Franklin Lakes, NJ, USA), according to the manufacturer’s protocol. The data acquisition and analysis were done with a FACS Canto II flow cytometer and data were acquired and analyzed using Diva software (Becton Dickinson, Franklin Lakes, NJ, USA).

#### 2.4.5. Antibodies

The primary antibodies used for western blot analyses are the following: mouse monoclonal anti-human β-actin (Clone AC-15; dilution 1:5000) from Sigma Aldrich (St. Louis, MO, USA); rabbit monoclonal anti-human Cleaved poly (ADP-ribose) polymerase (c-PARP) (Clone Asp 214; dilution 1:1000); rabbit monoclonal phospho-Akt (Ser473) (clone D9E, dilution 1.500); rabbit polyclonal phospho-mTOR (Ser2448) (dilution 1:500); rabbit polyclonal phospho-4E-BP1 (Ser65) (dilution 1:500); rabbit monoclonal Akt1 (clone D9R8K, dilution 1:500); rabbit polyclonal 4E-BP1 (dilution 1:500); and rabbit polyclonal mTOR (dilution 1:500), all from Cell Signaling (Cell Signaling Technology, Inc., Danvers, MA, USA).

#### 2.4.6. Statistical Analysis

All the experiments were performed at least in triplicate, as indicated, with technical repetition when possible. The results are expressed as mean ± standard error of the mean (SEM). A one- or two-tailed paired Student *t*-test with a normal-based 95% CI was applied for statistical analysis, as indicated. Statistical significance was defined as *p* < 0.05.

### 2.5. Validation Assays to Confirm T126 as a True Hit

#### 2.5.1. Intrinsic Fluorescence Assay

Compound **T126** was serially diluted in 384-well black plates with Hepes 100 mM, DMSO 4%, Brij 0.003%, Tween 20 0.004%, MgCl_2_ 10 mM, at a pH = 7.5 and incubated for 2 h at 37 °C. After the incubation time the intensity of fluorescence (λexc: 485 nm; λemm: 535 nm) was detected in a Tecan M1000 Pro.

#### 2.5.2. Solubility Screen Assay to Determine Chemical Aggregation

The stock solutions (10–2 M) of the compound **T126** were diluted to a decreased molarity, from 100 µM to 0.0001 µM, in 384-well transparent plates (Greiner 781801) with 1% DMSO: 99% buffer Hepes 100 mM, DMSO 4%, Brij 0.003%, Tween 20 0.004%, MgCl_2_ 10 mM, at a pH = 7.5. Incubation occurred at 37 °C and the results were read after 2 h in a NEPHELOstar Plus (BMG LABTECH, Ortenberg, Germany). The results were adjusted to a segmented regression to obtain the maximum concentration in which the compounds were soluble.

#### 2.5.3. UV-VIS Titration Assay with an Increasing Amount of MgCl_2_

The potential ability of **T126** to chelate Mg^2+^ ions was evaluated by spectrophotometric methods using a T70+ UV-Vis Spectrometer (PG Instruments, Leicestershire, UK) and a quartz cuvette with a 1 cm optical path. MgCl_2_. 5H_2_O 1M solution was prepared from powder (VWR International, Milan, Italy). The compound **T126** was dissolved in DMSO to a final concentration of 10 mM and diluted with a milliQ water/DMSO mixture to a final concentration of 100 µM (4% DMSO). The obtained solution was placed in a cuvette and the UV-Vis spectra was recorded between 200 and 500 nm using water milliQ (with 4% DMSO) as reference. Thereafter, small volumes of MgCl_2_ aqueous solution were added both in the sample and in the reference cuvettes to obtain a series of solutions containing increasing concentrations of MgCl_2_ (1.25, 2.5, 5 and 10 mM), carefully pipetted for mixing, and the UV-Vis spectra were repeated. Each experiment was performed in triplicate.

## 3. Results and Discussion

### 3.1. Computer-Aided Identification of Novel AKT1 Inhibitors

The idea of exploring AKT1 as a pharmacological target arose from serendipity during our efforts to discover new bioactive compounds against other protein kinases that were already the objects of our studies [40,41].

Among the investigated compounds, the in-house cyclohepta[*b*]thiophen-3-carboxyamide derivative **T187** (Figure 1) turned out to be inactive against the desired target (i.e., p38 MAPK), showing instead an unexpected inhibitory activity towards the catalytic domain of the AKT1 protein, with a percentage of enzymatic inhibition at 10 μM equal to 56%. This unforeseen result prompted us to better characterize the small molecule, which showed an IC_50_ = 11.4 ± 2.8 µM and a K_i_ = 4.19 ± 1.36 µM. Interestingly, a data literature search using SciFinder [42] pointed out that neither compound **T187** nor its chemical scaffold *N*-(5,6,7,8-tetrahydro-4*H*-cyclohepta[*b*]thiophen-2-yl)benzamide had ever been investigated in the context of AKT1 inhibition.

Intrigued by these results and stimulated by our interest and expertise in the field of kinases [43,44,45] and AML [46,47,48], we decided to take advantage of the serendipitous discovery of **T187** to rationally search for other AKT1 inhibitors within our in-house compound library (herein after called UNIPG-lib). Specifically, the UNIPG-lib is a chemical collection of ~3000 small molecules, including both target compounds and intermediates, synthesized and/or collected by the research team in the context of several drug discovery projects.

For our AKT1-targeted program, we carried out both ligand- and structure-based modeling studies to aid the selection of **T187**-like compounds for biological testing. First, the BioSolveIT Feature Trees (FTrees) [22] software tool was used to perform a fast 2D-similarity screening of the UNIPG-lib using **T187** as a query molecule. This ligand-based approach allowed us to select 247 available neighbors to the initial compound with a similarity threshold of at least 0.65. At this stage, to further rationalize the identification of potential AKT1 inhibitors, the collected compounds were submitted to molecular docking simulations to explore their binding potential to the investigated kinase. Within this aim, the RCSB PDB [23] was consulted to retrieve structural information on AKT1 and its ATP-site binders, highlighting that 28 crystal structures had been disclosed prior to December 2021. In particular, the most interesting co-crystallized ligands were present in the PDB structures 4EKL [25] and 4GV1 [24], showing the clinical trial inhibitors **ipatasertib** (GDC-0068) and **capivasertib** (AZD5363), respectively [20]. These small molecules shared a double interaction with the hinge region (residues Glu228 and Ala230) that anchored the ligand to the ATP-binding site (Appendix A). With such valuable structural information available, we decided to explore the ATP-binding pocket in the rational search of new potential AKT1 inhibitors. Among the two previously mentioned structures, 4GV1 showed the most favorable resolution and was thus selected as a protein model for molecular docking experiments of the previously cited 247 in-house compounds. Each molecule was docked using Glide [30] in the extra precision (XP) mode and the generated poses were then rescored with the MM-GBSA method within Prime [28]. The two co-crystallized advanced inhibitors were added to the ligand set for comparison, and the obtained binding poses were evaluated mainly based on the predicted intermolecular interactions, paying particular attention to the contacts that involved the key residues Ala230 and Glu228. The best docking conformations of **ipatasertib** and **capivasertib** agreed well with their experimental poses and in both cases the key interactions with the hinge residues were correctly reproduced (Appendix A).

Subsequently, the docking-based approach allowed the selection of 12 virtual hits as the most promising in-house AKT1 small-molecule binders (Appendix A), which were then evaluated for the anti-AKT1 activity.

In the employed biochemical assay, a compound was considered as a ‘primary active’ only when the inhibition of the AKT1 phosphorylation activity (hereinafter called %inh) was reduced by at least of 60%, employing a 10 μM compound concentration. Among the tested virtual hits, five compounds were thus selected as primary actives and evaluated in a dose–response assay to calculate the 50% inhibitory concentration (IC_50_) value (Figure 2 and Appendix A).

The [1,2,4]triazolo[1,5-*a*]pyrimidine derivative **T101** [35] and the cyclohepta[*b*]thiophene-3-carboxamide **T159** [32] were previously reported as allosteric inhibitors of HIV-1 reverse transcriptase-associated ribonuclease H. Analogously, compounds **T124**, **T126**, and **T128**, showing a common [4,5]thieno[2,3-*d*]pyrimidin-4(3*H*)-one moiety, were described as HIV-1 reverse transcriptase-associated ribonuclease H inhibitors by Masaoka et al. [34], and re-synthesized by us to serve as internal reference compounds.

Among them, compounds **T101**, **T126,** and **T159** exhibited an AKT1 inhibitory potency higher than **T187**, displaying IC_50_ values of 1.62 ± 0.15 µM, 1.99 ± 0.11 µM, and 2.12 ± 0.26 µM, respectively.

Notably, these three molecules turned out to be efficient inhibitors, as highlighted by the corresponding experimental values of LE. Particularly, **T159** and **T126** had an LE value ≥ 0.3 (Figure 2), which is the generally accepted lower limit of efficiency in a typical drug discovery program. It is worth noting that the 5,6,7,8-tetrahydrobenzo[4,5]thieno[2,3-*d*]pyrimidin-4(3*H*)-one derivative **T126** showed a LE value comparable to that of the clinical trial inhibitors **capivasertib** and **ipatasertib** (Appendix A). In addition, preliminary structure-activity relationships (SAR) analysis around this in-house compound can be deduced by examining the collected biological results (Appendix A). First, the replacement of the amide nitrogen atom in the tricyclic core with an oxygen atom seemed to be detrimental for kinase inhibition, as highlighted by comparing **T126** (%inh = 87%, IC_50_ = 1.99 ± 0.11 µM) and **T128** (%inh = 78%, IC_50_ = 26.6 ± 4.8 µM) with **T125** (%inh = 9%) and **T127** (%inh = 43%), respectively. This observation underlined that, although the in silico approach was successful in identifying new AKT1 inhibitors, the different impact on the biological activity of the pyrimidin-4(3*H*)-one moiety (**T126** and **T128**) compared to that of the oxazin-4-one (**T125** and **T127**) was not predicted correctly because similar docking poses and ligand-target interactions were proposed for the two series of compounds (Appendix A).

Therefore, we carried out molecular dynamics (MD) simulations to elucidate the molecular basis of the SAR data by overcoming the limitations of ordinary molecular docking studies, where the protein is typically treated as a rigid entity and no water molecules are included in the target preparation. The docking-generated AKT1–**T126** and AKT1-**T125** complexes were thus used as starting structures for three 200 ns MD-simulation replicas using Desmond software [31], and the analysis of the produced trajectories provided valuable clues to explain the SAR observation.

Specifically, the **T126** orientation generated from the docking study was well preserved during the entire execution time, with the overall ligand stability having been reached in the last 100 ns of the MD simulations (Appendix A). Interestingly, the nitrogen of the endocyclic amide established well-conserved, water-mediated interactions with the residues Thr291 and Asp292 (Appendix A and Figure 3).

Conversely, the three generated replicas for the inactive analog **T125** (Appendix A) did not detect common stable ligand-protein contacts, with the oxygen atom of the oxazin-4-one moiety never having been involved in significantly conserved intermolecular interactions (Appendix A).

As a second SAR insight, the size of the aliphatic ring of the tricyclic core appeared to be crucial to gain high inhibitory potency against AKT1. Indeed, the 6-membered ring of **T126** turned out to be the best substitution with respect to the 5-membered (**T128**) and 7-membered (**T124**, %inh = 60%, IC_50_ = 30.9 µM) systems.

In addition, it is well known that the level of interest in kinase targets has created a crowded chemical space landscape, rendering the discovery of novel chemical entities as kinase inhibitors challenging. Against this backdrop, compounds **T126**, **T159,** and **T101** were used as inputs for a structure search in SciFinder. To the best of our knowledge prior to December 2021, none of the three explored compounds had been reported in the literature, including publications and patents, as inhibitor of AKT1 or other kinases, as well as no information on the effect of these compounds on AML has emerged from our bibliographic survey.

### 3.2. Evaluation of the Identified AKT1 Inhibitors on AML Cell Lines

Encouraged by the variety of information gathered so far, we evaluated these compounds in cellular assays of AML models. Specifically, a cell viability assay was initially performed to determine the cellular IC_50_ of **T101**, **T126,** and **T159** on different human AML cell lines, namely, OCI-AML3, IMS-M2, OCI-AML2, MOLM-13, and SKM-1, which carry some of the most recurrent AML mutations [37,38]. We measured the ability of living cells to convert a redox dye (resazurin) into a fluorescent end-product (resorufin). A dose response curve was generated for each cell line using GraphPad Prism and the IC_50_ was established. While no interesting (IC_50_ higher than 30 µM) inhibitory activity was detected for **T101** and **T159** in all the explored cell lines, we obtained promising results for **T126** (Figure 4).

Indeed, **T126** IC_50_ values were 4.2, 4.3, 2.4, 9.2, and 6.9 µM for OCI-AML3, IMS-M2, OCI-AML2, MOLM-13, and SKM-1 cell lines, respectively (Figure 4A).

It is noteworthy that the low-micromolar inhibitory activities observed in the cell-based assays were similar to those obtained for the reference compound **NSC348884** [39] used as positive control (Appendix A), and were in keeping with the low-micromolar IC_50_ of **T126** on the isolated AKT1 protein (i.e., 1.99 µM).

We then investigated the effect of **T126** on OCI-AML3, IMS-M2, and OCI-AML2 cell growth (0–72 h), using a concentration of 5 µM, and found that **T126** was able to induce a significant growth inhibition at 48 and 72 h in all the three of the AML cell lines (Figure 4B).

To determine the effect of **T126** on the AKT1 intracellular mediated signaling pathway and cell death, we exposed our cell lines to an increasing concentration of **T126** and checked whether this small molecule acted upon their functional proteomic profiles. The effect of **T126** on the AKT signaling pathway was assessed by western blot analysis, focusing on the key targets. Upon increasing concentrations of **T126**, we observed at various levels in the different AML cell lines a decreased phosphorylation of AKT1 (phospho-AKT, at site S473) as an expression of the AKT1 activity inhibition [17], and of the downstream signaling molecule mTOR (phospho-mTOR, at site S2448). As a representative target of mTOR activity, we also analyzed the E4-BP1 protein and we noticed a reduction of its phosphorylation levels at site S65 (Figure 4C). As a marker of cell apoptosis, the apoptosis-related protein cleaved-PARP was detected upon treatment with **T126** 5 µM at 48 h (Figure 4C). The effect of **T126** on cell apoptosis was also measured by flow cytometry using Annexin V/7AAD probes for the identification of apoptotic cells. Notably, in accordance with western blot analysis, at the concentration of 5 µM, **T126** induced a significant level of apoptosis at 48 h in OCI-AML3, IMS-M2, and OCI-AML2 (Figure 4D), confirming in this setting, its anti-leukemic effect.

Finally, we performed a colony forming unit assay (CFU) using normal donor CD34+ hematopoietic stem/progenitor cells (HSPC) which were exposed to escalating doses of **T126** (1, 2.5 and 5 µM) and plated in a Methocult medium for 14 days. Remarkably, at 1 and 2.5 µM, compound **T126** did not inhibit the colony forming ability of human CD34+ HSPC, suggesting that there was no toxicity at these lower doses. Importantly, whilst a slight inhibition emerged at the maximum dose of 5 µM, the clonogenic efficiency was not blocked, but only decreased by about 20% as compared to the untreated control (Appendix A). The obtained results were quite encouraging considering that the inhibition of p-AKT was already observed at an early time point (4 h) at the concentrations of 1 and 2.5 µM in the OCI-AML3 cell line (Appendix A), suggesting that the compound could also be used at lower doses in additional biological experiments. These results confirm the pivotal role of the AKT pathway in the maintenance of cell growth and survival of AML cells, as previously reported [4,5]. Moreover, AKT pathway activation has also been implicated as a mechanism of resistance to therapy, including novel agents [18,49], pointing to AKT1 as an attractive target for AML, more specifically by using AKT1 inhibitors in combination with other drugs. Considering this, it is worth noting that the three AKT isoforms are highly conserved and share about 80% of the amino acid sequence identity [50]. As a consequence, the majority of currently developed AKT inhibitors are pan-AKT inhibitors, and this concept is evident when considering the two AKT inhibitors mentioned earlier and undergoing clinical trials (i.e., **capivasertib** IC_50_-AKT1 = 3 nM, IC_50_-AKT2 = 8 nM, IC_50_-AKT3 = 8 nM, **ipatasertib** IC_50_-AKT1 = 5 nM, IC_50_-AKT2 = 18 nM, and IC_50_-AKT3 = 8 nM) [51]. Based on this, we are aware that **T126** may potentially also inhibit AKT2 and AKT3. However, in the AML cell lines used in this study, AKT1 is the most expressed isoform among the AKT family members. Indeed, the analyses of RNA seq data from these cell lines (https://cellmodelpassports.sanger.ac.uk/passports, accessed on 14 October 2022) clearly indicate that AKT1 shows a higher degree of expression compared to the other isoforms. In the light of this information and considering that the work herein reported falls within the early-stage drug discovery process, we have focused our efforts exclusively on AKT1 to identify a hit compound. Nevertheless, during the next planned hit-to-lead steps, biological characterization studies involving the other AKT family members will be also carried out.

### 3.3. Validation of T126 as a True Hit

To further validate **T126** as a promising hit, focused studies were conducted to rule out the possibility that the inhibitory potency detected in the AKT1 assays could be related to an artifact. Indeed, this small molecule possessed a catechol moiety that seemed to be important for AKT1 binding by establishing polar interactions (Figure 3). Although the presence of the catechol by itself did not guarantee a satisfactory potency against AKT1 as the moderately active (**T124** and **T128**) and inactive (**T125**, **T127,** and **T129**) derivatives shared this feature as well, it is well-known that the mentioned chemical group has been counted as one of the Pan-Assay Interference Compounds (PAINS) motifs [52,53].

Various mechanisms of assay interference or promiscuous behaviors have been reported as responsible for PAINS activity, including compound fluorescence effect, metal chelation, and chemical aggregation [52,53,54].

In the AKT1 inhibition assay, the enzymatic activity was evaluated by measuring the phosphorylation of a fluorescently-labelled peptide induced by the kinase. Indeed, the detection of the substrate peptide and the product phosphorylated peptide after a microfluidic mobility shift was performed by measuring the fluorescence given to the substrate and the product of the kinase reaction. Based on the methodology employed for quantifying the AKT1 activity, it was not expected that the intrinsic fluorescence of the compounds could interfere with the assay. Nevertheless, we determined the intrinsic fluorescence of **T126**, which emerged as not fluorescent at the excitation/emission wavelength (485–535 nm) used in the assay.

Second, since the AKT1 inhibition assay required MgCl_2_, the Mg-complexation ability has been investigated for the **T126** derivative to rule out any interference. Specifically, UV-vis spectra were recorded for the compound alone and in the presence of increasing concentrations of MgCl_2_. The results showed no shift in the maximum of absorbance (hypsochromic effect), nor any presence of an isosbestic point (Appendix A) up to 10 mM concentration of MgCl_2_ (as used in the biochemical assay), thus suggesting the inability of **T126** to chelate Mg^2+^ ions.

Finally, to discard the hypothesis of chemical aggregation, the presence of aggregates was evaluated in the same buffer employed in the AKT1 activity assay by a NEPHELOstar Plus (BMG LABTECH). Compound **T126** did not aggregate at concentrations up to 100 µM, thus rejecting that the AKT1 inhibition effect may be due to aggregation.

In conclusion, the 5,6,7,8-tetrahydrobenzo[4,5]thieno[2,3-*d*]pyrimidin-4(3*H*)-one derivative **T126** showed a clear anti-AKT1 effect and the growth inhibition and induction of apoptosis in AML cells at low micromolar concentrations, making it an appealing candidate for further testing in other biological assays. Furthermore, the low molecular weight (MW = 314,37) and promising LE value of **T126** render this small molecule an attractive starting point for future medicinal chemistry efforts directed at hit-to-lead optimization.

## Data Availability

Data are contained within the article or Appendix A.

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
