# Peer review of "From Serendipity to Rational Identification of the 5,6,7,8-Tetrahydrobenzo[4,5]thieno[2,3-d]pyrimidin-4(3H)-one Core as a New Chemotype of AKT1 Inhibitors for Acute Myeloid Leukemia"

_pharmaceutics, 2022, doi:10.3390/pharmaceutics14112295_

Round 1

Reviewer 1 Report

This a well written and excellent paper dealing with the in silico conception of AKT1 inhibitor for acute leukemia treatment leading to the identification of a new hit which will serve as starting point in future medicinal chemistry study for lead optimization. The hit was tested on pertinent cell lin models of AML representative of wild types and mutated AML. The rational and the design of the experiments is correct. the data fit with conclusions from the authors

minor points:

line 189 : replace "in 1 M solution" by "in a 1M solution"

line 189-190 in which solvent was solubilized the precipitate?

what is the purity of compound T187

line 209 replace "Mentzen" by "Menten"

line 2010 specify % v/v or w/v

line 219 : " were previously reported" provide a reference

line 227 "the appropriate plating density" this expression is more convenient for adherent cells

line 229-230 249 250 and maybe other lines: change 106 and 105 by 106 and 105

line 237: replace "Cell Titer" by "CellTiter"

line 272 change 1,5 by 1.5, Ph by pH, Glycerol by glycerol and Bromophenol by bromophenol

Reviewer 2 Report

1. I do not have a background in the computational or chemistry synthesis field, so I am unable to judge those related contents.

2. It will be helpful to have a positive control for the biological assays if possible, to show the drug would really work in killing the cancer cells in vitro. And the efficacy and potency might be shown by referring to the positive control. Some in vivo animal models will be very helpful.

3. In figure 4A, the dose-responsive curve is inconsistent with the description in the text. The curve indicates that a >10M concentration of the compound is needed to completely inhibit cell growth. Whereas, it says less than 10uM was needed.

4. Figure 4 needs a title.

5. Figure 4C should include the bands of total protein for each phosphorylated form. Different time points for the same protein subject on the same cell line should be blotted in the same blot.

6. The entire blot for each WB experiment should be included in the supplemental information. 

7. Introduction and discussion should include the rationale and interpretation of those protein targets in figure 4.

Reviewer 3 Report

This is an interesting paper showing findings on T126 as a potential of AKT1 inhibitors as anti-AML agent. With a few small amendments I recommend this work for publication.

My concern is with the introduction of the paper which seems a little short and might need further contextualization.

I think authors need to make it clearer the AKT1 role in AML as there are multiple studies suggesting a paradoxical role of AKT1 in other cancer type by inhibiting certain tumor cell migration or otherwise in other tumors.

 Do the authors think these compounds may affect other AKT family members and discuss the potential impact, and shortcomings? These may be included as a part of discussion of results to improve the quality of presentation.

Round 2

Reviewer 2 Report

p70 S6K was mentioned in the introduction more than once, but not bloted with other kinases in figure 4. This can be improved.

Author Response

We thank the reviewer for this comment.

As mentioned in the Introduction, p70S6K and 4E-BP1 are both downstream targets of mTOR, which in turn is a downstream target of AKT.

In our manuscript, we have analyzed one of these proteins, i.e. E4-BP1, as the representative target of the mTOR activity to prove the inhibitory effect of our hit compound T126 on the AKT downstream pathways (and we have now rephrased one sentence in the “Results and Discussion” to make this concept clearer).

Thus, in this context, we believe that the inclusion of p70S6K in our analysis would not be particularly relevant to the main objectives of the submitted research work. Furthermore, the antibodies for the total and phosphorylated forms of p70S6K are not currently available in our lab and their acquisition and related biological assays would considerably delay the revision of this manuscript.

However, we thank the reviewer again for this constructive comment which we will consider to expand the biological characterization of our compounds in the following hit-to-lead process.